# Evidence of Zika virus circulation in asymptomatic pregnant women in Northeast, Brazil

**Rebeca Costa Castelo Branco**[1], **Patrícia Brasil**[2], **Josélio Maria Galvão Araújo**[3], **Flávia Oliveira Cardoso**[2]\*, **Zulmira Silva Batista**[1], **Valéria Maria Souza Leitão**[4], **Marcos Antonio Custódio Neto da Silva**[5], **Lailson Oliveira de Castro**[6], **Joanna Gardel Valverde**[7], **Selma Maria Bezerra Jeronimo**[7], **Josélia Alencar Lima**[1], **Raimunda Ribeiro da Silva**[8], **Maria do Carmo Lacerda Barbosa**[9], **Luciane Maria Oliveira Brito**[1], **Marcelo Antônio Pascoal Xavier**[10], **Maria do Desterro Soares Brandão Nascimento**[1]\*

1 Post-Graduate Program in Adult Health, Federal University of Maranhão, São Luís, Maranhão, Brazil, 2 Oswaldo Cruz Foundation (FIOCRUZ), Rio de Janeiro, Rio de Janeiro, Brazil, 3 Microbiology and Parasitology Department, Federal University of Rio Grande do Norte (UFRN), Natal, Rio Grande do Norte, Brazil, 4 Medicine III Department, Federal University of Maranhão, São Luís, Maranhão, Brazil, 5 Post-Graduate Program of Internal Medicine, State University of Campinas, Campinas, São Paulo, Brazil, 6 Medicine Faculty, Federal University of Maranhão, São Luís, Maranhão, Brazil, 7 Biochemistry Department and Institute of Tropical Medicine of Rio Grande do Norte, Federal University of Rio Grande do Norte (UFRN), Natal, Rio Grande do Norte, Brazil, 8 Pathology Department, Federal University of Maranhão, São Luís, Maranhão, Brazil, 9 Medicine I Department, Federal University of Maranhão, São Luís, Maranhão, Brazil, 10 Oswaldo Cruz Foundation, Belo Horizonte, Minas Gerais, Brazil

\* flaviaoc09@gmail.com (FOC); cnsd_ma@uol.com.br (MDSBN)

## Abstract

### Background

Zika virus (ZIKV) is a flavivirus associated with microcephaly and other fetal anormalities. However, evidence of asymptomatic ZIKV infection in pregnant women is still scarce. This study investigated the prevalence of Zika infection in asymptomatic pregnant women attending two public maternities in Maranhão state, Northeast Brazil.

### Methods

A total of 196 women were recruited at the time of delivery by convenience sampling from two maternity clinics in São Luís, Maranhão, Brazil, between April 2017 and June 2018. Venous blood, umbilical cord blood and placental fragments from maternal and fetal sides were collected from each subject. ZIKV infection was determined by reverse transcription polymerase chain reaction (RT-qPCR) for ZIKV and by serology (IgM and IgG). Nonspecific laboratory profiles (TORCH screen) were obtained from medical records.

### Results

The participants were mostly from São Luís and were of 19–35 years of age. They had 10–15 years of schooling and they were of mixed race, married, and Catholic. ZIKV was identified in three umbilical cord samples and in nine placental fragments. Mothers with positive ZIKV RT-qPCR were in the age group older than 19 years. Of the 196 women tested by

**Data Availability Statement:** All relevant data are within the manuscript and its Supporting Information files.

**Funding:** This work was supported by the Foundation for the Support of Research and Scientific and Technological Development of Maranhão – FAPEMA and Health Minister from Brazil - PPSUS [grant number 008/2016 - MDSBN] www.fapema.br/www.capes.gov.br and Coordination for the Improvement of Higher Education Personnel (CAPES) – [grant number 21/2018 – Notice Legal Amazonia] · that financed the materials needed for the execution of this research. The funders had no role in study design, data collection and analysis, decision to publish, or preparation of the manuscript.

**Competing interests:** The authors have declared that no competing interests exist.

ZIKV rapid test, 6 and 117 women were positive for anti-ZIKV IgM and anti-ZIKV IgG antibodies, respectively. Placental Immunohistochemistry study detected ZIKV in all samples positive by RT-PCR. The newborns did not show any morphological and/or psychomotor abnormalities at birth.

## Conclusions

Asymptomatic ZIKV infection is frequent, but it was not associated to morphological and/or psychomotor abnormalities in the newborns up to 6 months post-birth. Although pathological abnormalities were not observed at birth, we cannot rule out the long term impact of apparent asymptomatic congenital ZIKV infection.

## Author summary

Zika virus (ZIKV) belongs to family Flaviviridae, genus Flavivirus and it is transmitted by the bite of female Aedes mosquitoes. In October 2015 an unexpected increase in the number of newborns with microcephaly in Brazil. After investigation, Zika virus was strongly related to microcephaly. Considering the epidemiological importance of ZIKV infection during pregnancy and its repercussions on the mother-fetus pair, the aim of the present study was to investigate ZIKV infection in the placenta and umbilical cord blood of women served by the Brazilian Health System in São Luís, state of Maranhão, Northeast, Brazil, after the first ZIKV epidemic. This study investigated the prevalence of Zika infection in asymptomatic pregnant women attending two public maternities in Maranhão state, Northeast Brazil. We believe that these findings will contribute to the need of continuous awareness of the risk of ZIKV infection in pregnancy and the need to improving the health care and strategic planning of public policies regarding obstetric and neonatal care.

## Introduction

Zika virus (ZIKV) belongs to family *Flaviviridae*, genus *Flavivirus* and it is transmitted by the bite of female *Aedes* mosquitoes. It was first described in Africa in 1947 in samples isolated from *Rhesus* monkeys and later from humans [1]. The first confirmed case in the Americas was reported in May 2015 [2]; however, phylogenetic and molecular analyses indicated that it was introduced in Brazil between May and December 2013 [3–5]. In French Polynesia, in 2013 and 2014, approximately 30,000 cases of ZIKV fever were recorded, most of them with mild symptoms. However, a marked increase in the number of Guillain-Barré syndrome which was associated with ZIKV infection was reported [6].

In October 2015, an unexpected increase in the number of microcephaly in newborns was observed in Brazil, initially in the state of Pernambuco and subsequently in other Brazilian states. This increase occurred after the introduction of ZIKV and this virus was isolated from fetus. Later other places as French Polynesia analyzed retrospectively their data for newborns and identified also an increase in microcephaly post-introduction of ZIKV. The outbreak of microcephaly associated with ZIKV infection lasted for almost a year in Brazil. ZIKV has exhibited continued transmission in a large part of the Brazilian territory, with no apparent increase in microcephaly newer cases [7].

The term congenital Zika syndrome describes a series of congenital defects associated with ZIKV infection, including microcephaly, complex brain malformations, and eye lesions. Although microcephaly is a classic finding for its diagnosis, there is evidence that subtle but destructive brain and ocular injuries can also occur in infants with normal head size at birth [8].

Although vector transmission predominates in ZIKV infections in humans, transmission through sexual contact [9–11], blood transfusion [12,13], occupational exposure in laboratory workers, and intrauterine and intra-birth transmission [14] have also been reported. Asymptomatic infection correspond an up to 80% of ZIKV infections [15].

Considering the epidemiological importance of ZIKV infection during pregnancy and its repercussions on the mother-fetus pair, the aim of the present study was to investigate the prevalence of Zika infection in asymptomatic pregnant women attending two public maternities in Maranhão state, Northeast Brazil, after the first ZIKV epidemic. This study investigated the prevalence of Zika infection in asymptomatic pregnant women in Maranhão state, Northeast Brazil. We believe that these findings will contribute to the need of continuous awareness of the risk of ZIKV infection in pregnancy and the need to improving the health care and strategic planning of public policies regarding obstetric and neonatal care.

## Methods

### Ethics statement

The protocol was revised and approved by the Ethical Research Committee of the University Hospital of the Federal University of Maranhão (number 2.475.441). All participants provided an Informed Consent Term.

All participants provided a written formal consent.

### Study design and population

This was a cross-sectional study carried out at the Maternity and Infant Care Unit of the University Hospital of the Federal University of Maranhão (HU-UFMA) and the State Maternity Benedito Leite in São Luís, Maranhão, Brazil between April 2017 and June 2018. The study participants were recruited by contingency sampling. The inclusion criteria were women in active labor or undergoing elective Cesarean sections and agreed to participate in the study. Participants who had positive serology for syphilis, toxoplasmosis (immunoglobin M [IgM]), rubeola (IgM), human immunodeficiency virus (HIV), hepatitis B (hepatitis B surface antigen [HBsAg]), or cytomegalovirus (IgM); and those who presented a compromised placenta due to intrauterine growth restriction, polyhydramnios, oligohydramnios, pre-eclampsia, antiphospholipid antibody syndrome were excluded. Women with clinical symptoms of ZIKV infection were also excluded. The size of the study population was estimated based on the occurrence of ZIKV infection in the state of Maranhão during the first wave.

The number of participants recruited considered the prevalence of estimated ZIKV infection in Maranhão, Brazil. A total of 203 pregnant was recruited, 7 were excluded because of difficulties in blood collection. A final sample of 196 pregnant women from the two maternities was then studied.

### Data collection

ZIKV infection was defined by laboratorial results (positivity of rapid test, ELISA and PCR) and exclusion of other arboviruses during the period of the study. Sample collection was performed between January 2017 and June 2018. An interview was performed at the time of

recruitment when a structured questionnaire that included information about sociodemographic characteristics, sexual habits, reproductive history, clinical and obstetric history was collected. Data for the current pregnancy was obtained from the prenatal card.

Participants underwent an obstetric examination performed by the assistance team. Peripheral venous blood was collected from the antecubital vein (2–5 mL). Umbilical cord blood was collected during placental expulsion and three pairs of placental tissue samples from maternal and fetal sides were collected close to the umbilical cord insertion and placental apices after placental expulsion.

Placental sample pairs were placed in separate jars. One sample was fixed in 10% formalin for subsequent processing and histopathological evaluation. The other sample was placed in RNA*later* Stabilization Solution (Invitrogen, USA), stored at 4˚C overnight, and stored at -80˚C for subsequent virus detection by RT-qPCR. RNA extraction was performed at the Multiuser Laboratory of the Post-graduate Program on Adult Health, at the premises of the Tumor and DNA Bank of Maranhão.

### Infant clinical assessment

Anthropometric measures at birth (weight, length, and head circumference) were obtained for all newborns. Physical examinations were performed for all infants by the same pediatricians using a standard form. Microcephaly was defined as a head-circumference z score of less than −2 (moderate) or less than −3 (severe) according to INTERGROWTH 21st. Small-for-gestational-age infants were defined as infants with body-weight z scores of less than −1.28 at birth.

### Testing for arbovirus infections

**Rapid test.** All sera samples (n = 196) were screened for ZIKV infection using the rapid test (TR DPP Zika IgM/IgG, Bio-Manguinhos, Oswaldo Cruz Foundation, Rio de Janeiro, RJ, Brazil) This immunochromatographic test is double-path platform and evaluates presence of IgM and IgG antibodies with sensitivity and specificity around 95%. The test is negative for IgM and IgG for values of 0–22, indeterminate for values of 23–27, and positive for values equal to or higher than 28.

**ELISA (ZIKV, Dengue and Chikungunya).** Serological evaluation to detect IgG anti-ZIKV, anti-DENV and anti-CHIKV was performed by ELISA and the protocol was based a previous study (Tsai et al., 2017) [16]. High-affinity COSTAR 3590 plates (Corning, USA) were coated with a total of 20 ng/well of Zika virus NS1 and four dengue virus NS1 serotypes and 25 ng/well of Chikungunya virus E2 antigen (Meridian Life Science, USA) in carbonate-bicarbonate buffer pH 9.6 at 4˚ C overnight, and subsequently blocked with 1% PBS-Tween buffer. Serum samples were diluted 1: 400 and placed on the plates in duplicate. HRP-conjugated anti-human IgG antibodies (Promega, USA) were added to the wells at a 1: 2000 dilution. The reaction was revealed by adding 2,2′-Azino-bis (3-ethylbenzthiazoline-6-sulfonic acid (ABTS) (Sigma-Aldrich, USA) and hydrogen peroxide (Sigma-Aldrich, USA). Reaction was stopped by adding a 5% sodium dodecyl sulfate (SDS) solution.

For each plate, three negative control samples and four positive control samples were used to validate the test. The cut-off of the reaction was calculated from the corrected mean of the negative controls plus three times the standard deviation. The indeterminacy zone was considered as the range of reading values between 10% of the cut-off value upwards or downwards. The reaction was considered valid when at least three of the four positive controls had valid positive results (above the indeterminacy zone). The readings were normalized for analysis by calculating the relative optical density, determined by the ratio of the sample's optical density to the respective cut-off (Relative optical density).

## Viral RNA extraction

Viral RNA was extracted from total umbilical cord blood samples using a QIAamp Viral RNA Mini Kit (Qiagen, USA) according to the manufacturer's recommendations. Placental fragments placed in RNA*later* Stabilization Solution were ground using a porcelain mortar and pestle and the homogenate was centrifuged for 15 min at 6000 RPM. The supernatant was collected and viral RNA was extracted using a QIAamp Viral RNA Mini Kit.

## RT-qPCR for ZIKV detection

ZIKV was detected by RT-qPCR (TaqMan), as described by Faye et al. (2013) [17]. Two forward and reverse consensus primers (Zika_qRT_F and Zika_qRT_R), complementary to the genes coding for the NS5 protein were used at a concentration of 200 nM, along with the probe (Zika_qRT_P) at a concentration of 125 nM (S1 Table).

RT-qPCR reactions were performed for each sample as follows: 2.5 μL of TaqMan FAST 1-Step Master Mix (2x) (Applied Biosystems, Foster City, USA), 0.2 μL of 200 nM Zika_qRT_F primer (Invitrogen, USA), 0.2 μL of 200 nM Zika_qRT_R primer (Invitrogen, USA), 0.2 μL of 125 nM Zika_qRT_P probe (Applied Biosystems, Foster City, USA), 1.9 μL of nuclease-free water (Promega, Madison, USA), and 5 μL of the extracted RNA were mixed and added to each well of a 96-well optical PCR plate (Applied Biosystems, Foster City, USA). The reactions were performed using an ABI Prism 7500 Fast thermocycler (Applied Biosystems, Foster City, USA). The PCR program consisted of an initial cycle at 50˚C for 5 min for reverse transcriptase activation, followed by enzyme inactivation and initial denaturation at 95˚C for 20 seconds, 40 cycles at 95˚C for 15 seconds for denaturation, primer annealing at 60˚C for 1 min, extension at 60˚C for 1 min, and final extension at 60˚C for 1 min.

## Histopathological and Immunohistochemistry placenta evaluation

Histological slides containing 5 μm sections of the paraffinized samples were incubated overnight, at 56˚C. Subsequently, the samples were subjected to dewaxing and rehydration steps, with three washes in xylol, for five minutes each, and three washes in ethyl alcohol (PA), for five minutes. After rehydration, heat-induced antigenic recovery was performed in 0.01 M sodium citrate solution (pH 6.0) at 90˚C for 20 minutes in steam and cooled to room temperature for 20 minutes. Endogenous tissue peroxidase and nonspecific proteins were blocked at different stages. Immunohistochemistry staining was performed with the primary anti-*Flavivirus* group antigen antibody [D1-4G2-4-15 (4G2)] (Absolute Antibody, INC). For detection of the primary antibody, the NovoLink Max Polymer Detection Novocastra TM kit (Leica Microsystems) was used. The sections were incubated with the NovoLink kit's universal polymer detection system for 30 minutes at room temperature. Subsequently, 200 μL of the developer solution provided by the kit containing the diamino-benzidine chromogen 3.3 (DAB) was added and the reaction incubated at room temperature for five minutes. Counter staining was performed by dipping the slides in Harris' hematoxylin solution (Code 248, Vetec) for 30 seconds. To control and monitor the quality of immunohistochemical reactions, external control (tissues from chicken embryos infected with ZIKV) was used.

## Statistical analysis

A digital databank was constructed using IBM SPSS Statistics for Windows, version 22.0. Data were transferred to a computerized databank using Epi-Info software for further analysis. A significance level of $p < 0.05$ and confidence interval of 95% were used. Variables with normal distributions were analyzed using parametric tests, while variables without normal distribution

were analyzed using non-parametric tests. Qualitative variables were analyzed using chi-square and Fisher's exact tests and odds ratios (ORs) (CI 95%). A multivariate analysis was performed using logistic regression model. Average, standard deviation, and minimum and maximum or median values were calculated for quantitative variables with normal distributions. The 25th and 75th percentiles were determined for variables with non-normal distributions. The association between epidemiological, clinical, and histopathological data and the laboratory results for ZIKV were evaluated by chi-square and Fisher's exact tests, with $p < 0.05$. All statistical analyses were performed using GraphPad Prism version 6 (GraphPad Software, CA, USA).

## Results

### Frequency of Zika RNA and antibody detection among biological samples

Among 196 pregnant women recruited we detected zika virus in 9 placental (4.6%), and 3 of them were also positive for umbilical cord blood. None of the positive cases present IgM positive thought Rapid test, but seven presented Positive IgG thought Rapid test. Only two presented IgG thought ELISA assay (Table 1).

### Anti- ZIKV, DENV and CHIKV antibodies

In order to detect others arbovirus antibodies 196 samples were tested to identify anti-CHIKV and/or anti-dengue 1–4 IgG. Antibodies against ZIKV NS1 protein were detected in 34 samples (17.3%), of which, 24 also presented antibodies against all four dengue serotypes. Forty four samples (22.3%) tested were negative for Zika and all dengue serotypes. Of the 196 samples tested, 59 presented antibodies against Chikungunya (30.1%) (Fig 1).

### Correlation of rapid test and ELISA

A total of 97.1% positive results in ELISA had also a positive rapid test results (Fig 2A). The sensitivity of the rapid test in relation to ELISA was 97%, whereas the specificity of the rapid test in relation to ELISA was 65%. In addition, a positive correlation (r = 0.4875; p <0.0001) was found between the results of the Biomanguinhos rapid test for Zika IgG detection and the relative optical density of the Zika ELISA for IgG detection against NS1 Protein (Fig 2B).

**Table 1. Results of molecular tests performed on samples from pregnant women exposed to Zika virus between April 2017 and June 2018, São Luís-Maranhão.**

| Variables | N | % |
|---|---|---|
| **RT-qPCR for umbilical cord** | | |
| Positive | 3 | 1.5 |
| Negative | 193 | 98.5 |
| **RT-qPCR for placenta** | | |
| Positive | 9 | 5.6 |
| Negative | 185 | 94.4 |
| Total | 196 | 100 |
| **Anti- ZIKV antibody** | | |
| Rapid test IgM positive | 6 | 3.0 |
| Rapid test IgG positive | 116 | 59.2 |
| ELISA IgG positive | 34 | 17.3 |

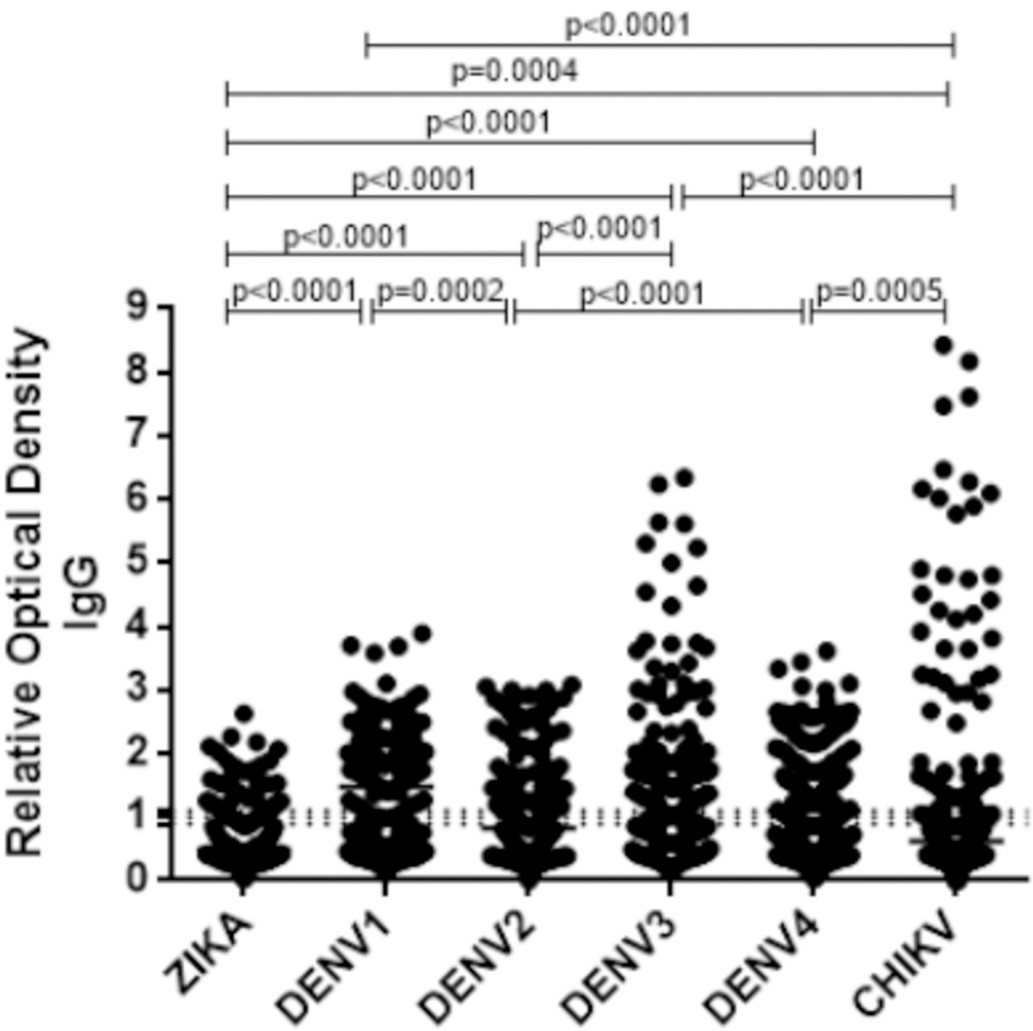

**Fig 1. Evaluation of the relative optical density of anti-NS1 IgG from Zika, Dengue (serotypes 1–4) and anti-E2 IgG from Chikungunya.** The interval between the dotted lines are values between 0.9 and 1.1 that were considered indeterminate.

### RT-PCR

**Placenta histopathological evaluation.**    Nine (4.6%) placentas analyzed, presented the ZIKV genome detected by qRT-PCR.

Fig 3 presents the anatomopathological results. The most common findings by hematoxylin-eosin (HE) were placental hypoplasia (nine), accelerated villous maturation (nine) and chronic villitis (nine). Other findings were acute intervilositis (three), chronic intervilositis (one), corangiosis (one), fibrina deposition (one), bebbing (one), cellular fusion (one) and acute chorioamninonitis (one) (Fig 3).

**Placenta immunohistochemistry.**    In order to confirm the ZIKV presence in placenta samples, immunohistochemistry was performed with anti-ZIKV antibodies. ZIKV was evidenced in the 9 samples tested (Fig 4).

**Sociodemographic data.**    Of the 196 pregnant women recruited, 63.3% resided in the city of São Luís, 85.2% were 19–35 years of age, 66.8% had 10–15 years of schooling, 64.3% were of mixed race, 69.9% were married and 61.2% were catholic (Table 2).

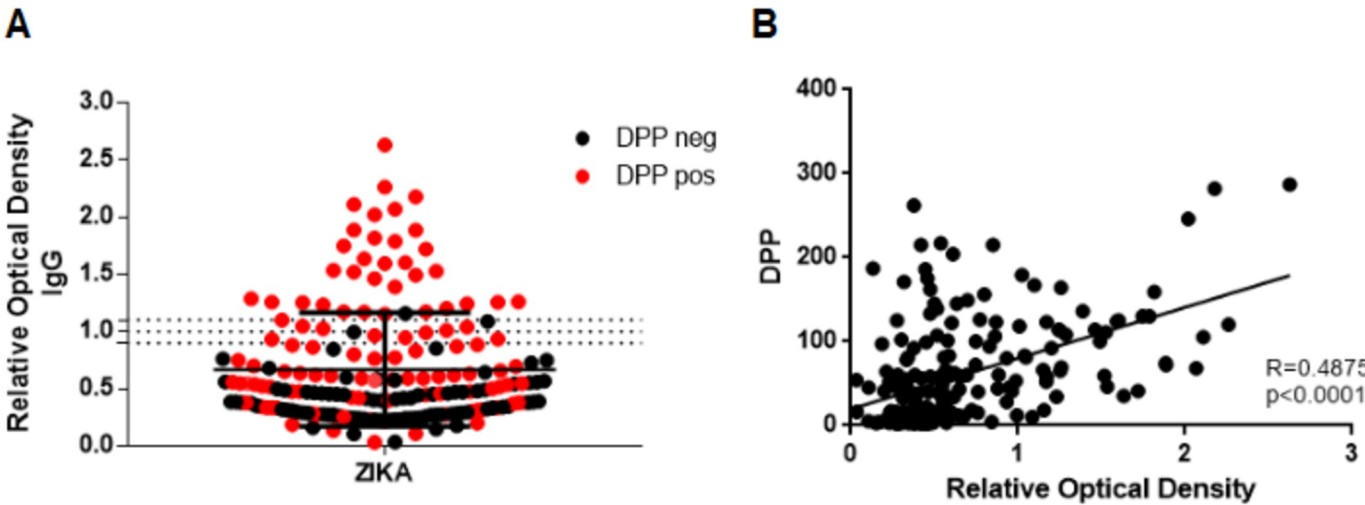

**Fig 2.** (A) Evaluation of the relative optical density of anti-NS1 Zika IgG for individuals with rapid IgG test negative (black) and positive (red). (B) Correlation between the Zika IgG rapid test result and the relative optical density of the Zika anti-NS1 IgG ELISA.

**Morphologic and or/psychomotor changes in newborns.** Tables 3 and 4 evidenced the crossing of newborn variables with respect to the results of the RT-PCR for umbilical cord and placenta. Gestational age < 37 weeks at delivery was associated to the presence of ZIKV in the umbilical cord detected by RT-PCR (p<0.05). The mean head size at birth in both groups was the same (34.5).

No association was observed between the dissemination of ZIKV in the placenta and umbilical cord and the presence of fetal abnormalities.

## Discussion

This is the first report of laboratory confirmed ZIKV infection in asymptomatic pregnant women in Maranhão, Brazil. Our results revealed a predominance of pregnant women

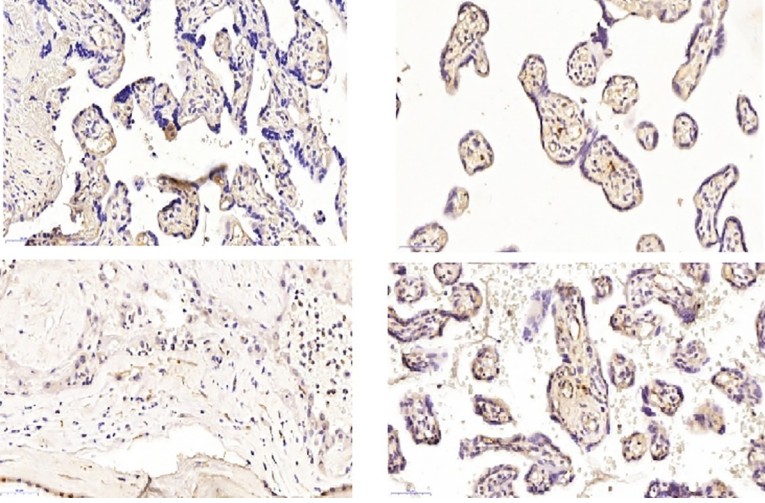

**Fig 3. Representative evaluation of placental, hematoxylin–eosin findings.** (A) granuloma (40 × objective) (B) leukocyte infiltrate (40 × objective); (C) acute chorioamninonitis (40 × objective); (D) Hofbauer cells (40 × objective).

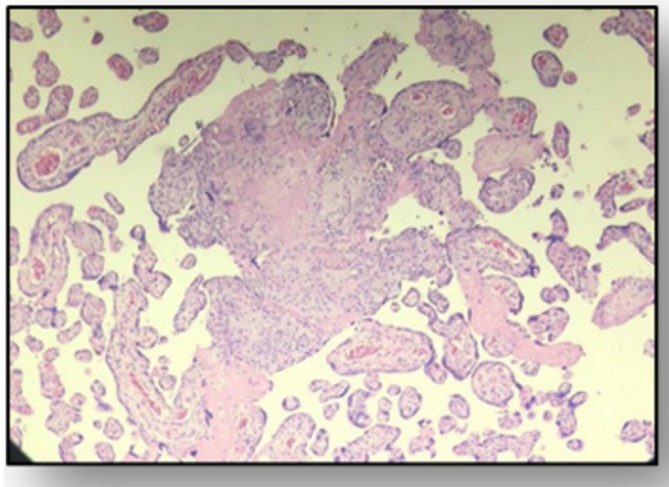
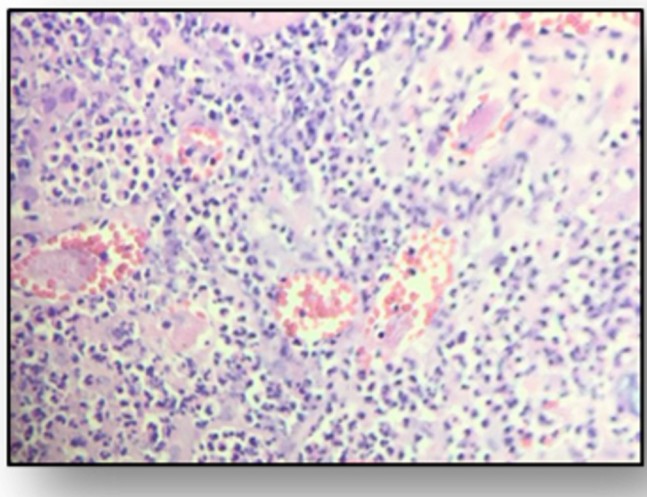
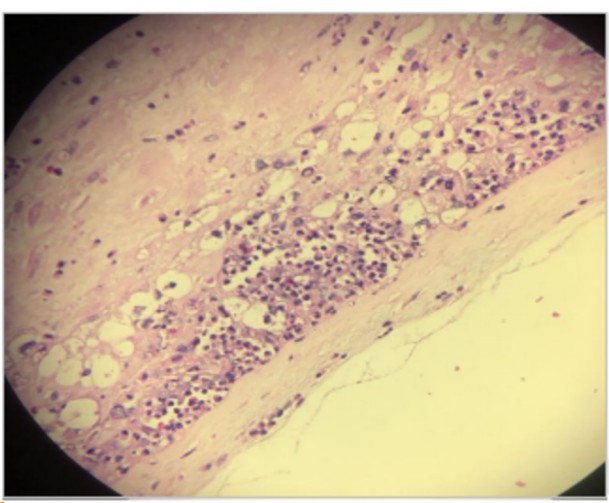
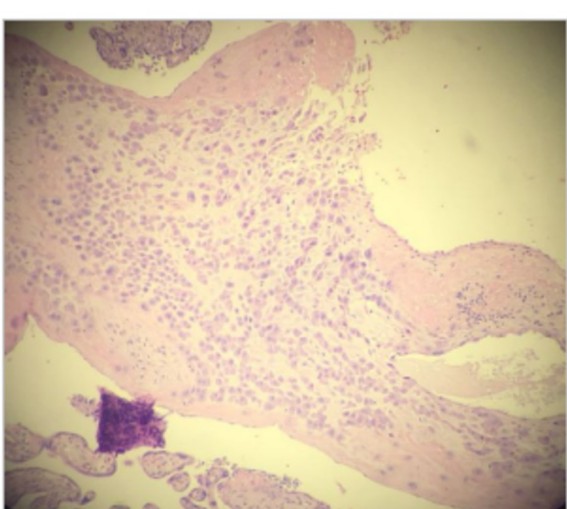

**Fig 4. Photomicrography of the placental samples immunostained with anti-ZIKV MAb and stained with Harris's hematoxylin (squares).** (A) Case 1. Placenta sample (chorion frondosum) showing placental hypoplasia, accelerated villous maturation and chronic villitis (TORCH-like). (B) Case 2—Placenta sample showing accelerated villous maturation and corangiosis. (C) Case 3—Placenta sample showing accelerated villous maturation and acute chorioamninonitis. (D) Case 4- Placenta sample showing accelerated villous maturation and fibrin deposition.

between 19 and 35 years of age (84.8% of the participants); 10.7%, 18yo or less than 18 yo and 4.6% older than 36 years of age. This distribution of age towards the old ones (>35yo) was different from that reported (6.84%) by the World Health Organization (2018) [18]. Most pregnant women in the study were not in the risk group for obstetric complications (<15 yo and > 35 yo), according to the Ministry of Health of Brazil [19].

Regarding education, 67% of the pregnant women in the present study had between 10 and 15 years of schooling and, therefore, a good level of education which may lead to the presumption of optimal/good prenatal care. Our data are in contrast with that reported by Gonzaga et al. (2016) [20], who evaluated prenatal care and associated risk factors and reported that 51% of participants had up to 8 years of schooling. A low level of education place pregnant

**Table 2. Sociodemographic characteristics of the study participants.**

| Variables | N | % |
|---|---|---|
| **Origin** | | |
| São Luís | 124 | 63.3 |
| Other municipalities | 62 | 36.7 |
| **Education (years)** | | |
| 0–4 | 01 | 0.6 |
| 5–9 | 64 | 32.6 |
| 10–15 | 131 | 66.8 |
| **Age (years)** | | |
| ≤18 | 21 | 10.7 |
| 19–35 | 167 | 85.2 |
| ≥36 | 08 | 4.1 |
| **Marital status** | | |
| Single | 59 | 30.1 |
| Marital union | 137 | 69.9 |
| **Race** | | |
| White | 22 | 11.2 |
| Black | 48 | 24.5 |
| Mixed | 126 | 64.3 |
| **Number of children** | | |
| 0 | 20 | 10.2 |
| 1–2 | 150 | 76.5 |
| 3–4 | 17 | 8.7 |
| 5 | 09 | 4.6 |
| **Place of residence** | | |
| Rural area | 15 | 7.7 |
| Urban area | 181 | 92.3 |
| **Current religion** | | |
| Catholic | 120 | 61.2 |
| Evangelical | 45 | 22.9 |
| None | 22 | 11.3 |
| Other | 09 | 4.6 |
| **Total** | **196** | **100** |

women at a disadvantage in terms of material resources and social support, which becomes an obstetric risk factor. This contrast may be explained by the fact that our study was performed in the state capital, from which most of the subjects originated.

Study conduct by Brasil and colleagues (2016) [21] in Rio de Janeiro evidenced that 182 women from a total of 345 women tested positive for ZIKV in blood, urine, or both. Infection with chikungunya virus was identified in 42% of women without ZIKV infection versus 3% of women with ZIKV infection. The positivity of this study was that pregnant women enrolled had a history of rash and in our study they were completely asymptomatic.

ZIKV was identified in placenta samples from 9 different women and in three of 197 umbilical cord blood samples. Our main histological findings by HE were placental hypoplasia, accelerated villous maturation and chronic villitis, where there is usually a higher concentration of ZIKV, and the highest concentration of Hofbauer cells (HCs) [22]. We also evidenced hyperplasia of HCs in the placenta of infected women which can contribute to the spread of ZIKV and to promote the vertical transmission [23,24] or fetal disruption, as reported by

**Table 3. Relationship for umbilical cord RT-PCR and newborn variables.**

| Variables | | Positive | Negative | Total | p-value |
|---|---|---|---|---|---|
| | | **3** | **193** | **196** | |
| Sex | Female | 1 | 102 | 103 | 0.6065 |
| | Male | 2 | 91 | 93 | |
| Apgar 1˚ Min | < 8 | 1 | 28 | 29 | 0.3814 |
| | ≥ 8 | 2 | 165 | 167 | |
| Apgar 5˚ Min | < 8 | 0 | 4 | 4 | 1.0000 |
| | ≥ 8 | 3 | 189 | 192 | |
| Gestational age at delivery | < 37 weeks | 1 | 2 | 3 | **0.0494** |
| | 37–41 weeks and 6 days | 2 | 183 | 185 | |
| | ≥ 42 weeks | 0 | 8 | 8 | |
| Weight | Median | 3030.0 | 3300.0 | | |
| | Amplitude | 760.0 | 2205.0 | | |
| Cephalic perimeter | Median | 34.5 | 34.5 | | |
| | Amplitude | 1.0 | 7.0 | | |

Beaufrère et a (2019) [25]. Umbilical cord a region with the lowest cellularity, presented lower concentration of ZIKV, as expected.

Besides hyperplasia of Hofbauer cells, a result of delayed villous maturation with additional stromal changes and the most prevalent alteration in our placenta samples, we also detected chorionic villi with calcification, fibrosis, perivillous fibrin deposition, patchy intervillositis and focal villitis in the samples studied, found also in other studies [22,26–28]. Venceslau et al (2020) [22] also evidenced, maternal vascular malperfusion, placental hypoplasia, and maternal–fetal hemorrhage (intervillous thrombi) as the most common morphological and anatomopathological findings.

Immunohistochemical (IHC) analysis of the placental tissue samples using anti-flavivirus MAb confirmed the presence of ZIKV in all placenta samples that were positive by RT-PCR.

Differently from other studies [21] none of the newborns from positive placenta and umbilical cord samples evidenced morphological or psychomotor changes at delivery. It is not clear

**Table 4. Relationship for placenta RT-PCR results and newborn variables.**

| Variables | | Positive | Negative | Total | p-value |
|---|---|---|---|---|---|
| | | **9** | **187** | **196** | |
| Sex | Female | 4 | 98 | 102 | 0.7391 |
| | Male | 5 | 89 | 94 | |
| Apgar 1˚ Min | < 8 | 2 | 27 | 29 | 0.6233 |
| | ≥ 8 | 7 | 160 | 167 | |
| Apgar 5˚ Min | < 8 | 0 | 4 | 4 | 1.0000 |
| | ≥ 8 | 9 | 183 | 192 | |
| Gestational age at delivery | < 37 weeks | 1 | 2 | 3 | 0.1719 |
| | 37–41 weeks and 6 days | 8 | 177 | 185 | |
| | ≥ 42 weeks | 0 | 8 | 8 | |
| Weight | Median | 3235.0 | 3300.0 | | |
| | Amplitude | 1220.0 | 2205.0 | | |
| Cephalic perimeter | Median | 34.5 | 34.5 | | |
| | Amplitude | 3.6 | 7.0 | | |

whether placental alterations are related to fetal adverse or neonatal outcomes or if they are associated with maternal–fetal transmission of ZIKV [29]. Further studies need to be performed to correlate this alterations with morphological changes in newborns.

According to studies by Noronha et al. (2016) [30] and Singh et al. (2016) [31], most infants with congenital zika syndrome were children born from symptomatic women in the first trimester, which can explain why the newborns of our study did not present morphological alterations, such as microcephaly.

The main strengths of this study is that we used the reference pattern for diagnosis (RT-qPCR) of ZIKV infection and we were able to study and confirm the pathological aspects and the virological damages of placenta infection by the ZIKV, even though the pregnant women were asymptomatic.

The main limitation of the study was the low prevalence of ZIKV, and the small sample size of women with laboratory evidence of ZIKV infection during pregnancy, which limits the power of the study to detect morphological and/or psychomotor abnormalities in the newborns.

## Conclusions

This is the first report of ZIKV infection of asymptomatic pregnant women in São Luís, Maranhão, Brazil. The study results showed the presence of ZIKV in the placenta and umbilical cord of women without clinical manifestations. Despite the presence of ZIKV in umbilical cord blood, it was not associated to morphological and/or psychomotor abnormalities in the newborns.

## Supporting information

**S1 Table. Primers used in RT-qPCR for ZIKV detection (Y = T or C, R = A or G).**
(DOCX)

## Acknowledgments

We are grateful to the University Hospital from Federal University of Maranhão and Benedito Leito Maternity for providing the patients that were included in this research.

We are grateful to Federal University of Rio Grande do Norte and Oswaldo Cruz Foundation for the development of experimental tests.

We would like to thank Dr. Marco Antonio Carneiro Menezes (Vice President of Environment, Attention and Health Promotion—FIOCRUZ) for his help in acquiring the rapid tests for Zika and Bio-Manguinhos for supply the kits.

## Author Contributions

**Conceptualization:** Rebeca Costa Castelo Branco, Josélio Maria Galvão Araújo, Flávia Oliveira Cardoso, Zulmira Silva Batista, Maria do Desterro Soares Brandão Nascimento.

**Data curation:** Rebeca Costa Castelo Branco, Zulmira Silva Batista, Valéria Maria Souza Leitão, Lailson Oliveira de Castro, Maria do Desterro Soares Brandão Nascimento.

**Formal analysis:** Rebeca Costa Castelo Branco, Josélio Maria Galvão Araújo, Flávia Oliveira Cardoso, Zulmira Silva Batista, Valéria Maria Souza Leitão, Marcos Antonio Custódio Neto da Silva, Joanna Gardel Valverde, Selma Maria Bezerra Jeronimo, Josélia Alencar Lima, Maria do Carmo Lacerda Barbosa, Luciane Maria Oliveira Brito, Maria do Desterro Soares Brandão Nascimento.

**Funding acquisition:** Maria do Desterro Soares Brandão Nascimento.

**Investigation:** Rebeca Costa Castelo Branco, Josélio Maria Galvão Araújo, Flávia Oliveira Cardoso, Zulmira Silva Batista, Valéria Maria Souza Leitão, Lailson Oliveira de Castro, Joanna Gardel Valverde, Selma Maria Bezerra Jeronimo, Raimunda Ribeiro da Silva, Maria do Carmo Lacerda Barbosa, Marcelo Antônio Pascoal Xavier.

**Methodology:** Rebeca Costa Castelo Branco, Josélio Maria Galvão Araújo, Zulmira Silva Batista, Valéria Maria Souza Leitão, Joanna Gardel Valverde, Selma Maria Bezerra Jeronimo, Raimunda Ribeiro da Silva, Maria do Carmo Lacerda Barbosa, Marcelo Antônio Pascoal Xavier.

**Project administration:** Patrícia Brasil, Maria do Desterro Soares Brandão Nascimento.

**Resources:** Maria do Desterro Soares Brandão Nascimento.

**Supervision:** Josélia Alencar Lima, Luciane Maria Oliveira Brito, Maria do Desterro Soares Brandão Nascimento.

**Writing – original draft:** Rebeca Costa Castelo Branco, Marcos Antonio Custódio Neto da Silva.

**Writing – review & editing:** Rebeca Costa Castelo Branco, Patrícia Brasil, Marcos Antonio Custódio Neto da Silva, Luciane Maria Oliveira Brito, Maria do Desterro Soares Brandão Nascimento.

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
