## [Decision Letter · Decision Letter 0]

24 Sep 2020

Dear Dr Nascimento,

Thank you very much for submitting your manuscript "Evidence of Zika virus circulation in asymptomatic pregnant women in Northeast, Brazil" for consideration at PLOS Neglected Tropical Diseases. As with all papers reviewed by the journal, your manuscript was reviewed by members of the editorial board and by several independent reviewers. In light of the reviews (below this email), we would like to invite the resubmission of a significantly-revised version that takes into account the reviewers' comments. 

We cannot make any decision about publication until we have seen the revised manuscript and your response to the reviewers' comments. Your revised manuscript is also likely to be sent to reviewers for further evaluation.

Sincerely,

Abdallah M. Samy, PhD

Deputy Editor

Victor S. Santos

Deputy Editor

Editor Comments

Overall

The present manuscript will be strengthened by a grammar review with a native speaker.

Please, follow the PLoS Neglected Tropical Diseases guidelines.

Introduction

Please, edit the sentence: 

“This is the first study of ZIKV infection diagnosis in asymptomatic pregnant women in the state of Maranhão, Brazil, using different methods.” 

Please, use: “This study investigated the prevalence of Zika infection in asymptomatic pregnant women in Maranhão state, Northeast Brazil.

Methods

Please, inform the correct study design. For me, this was a cross-sectional study.

What the authors meant by: “Cases in which the tests could not be performed due to technical difficulties”. Please, clarify that sentence.

Provide a reference for the definition of microcephaly. Did the authors use INTERGROWTH 21st? If not used, this should be placed as a limitation of the study.

Provide more details about the rapid test used. Was a rapid diagnostic test based on antigen detection? Or was a rapid diagnostic test based on host antibody detection? What is the sensitivity and specificity of the test? Was it used as a screening test (test-to-test)? Or was it used for diagnosis? This must be taken into account.

The authors must include a section on the definition they used for ZIKV infection. Was ZIKV infection considered a positive result for any of the tests? Depending on the sensitivities and specificities of the tests, this may have led to an important bias.

Data analysis. Authors should consider providing more details about data analysis, as well as including a multivariate analysis.

Results

The authors showed data on other arboviruses. But there are no details on how they ran the tests in the Methods section. The same occurs with the presentation of anatomopathological results. 

Conclusion

Please, leave the conclusion in a paragraph.

Reviewer's Responses to Questions

**Key Review Criteria Required for Acceptance?**

**Methods**

-Are the objectives of the study clearly articulated with a clear testable hypothesis stated?

-Is the study design appropriate to address the stated objectives?

-Is the population clearly described and appropriate for the hypothesis being tested?

-Is the sample size sufficient to ensure adequate power to address the hypothesis being tested?

-Were correct statistical analysis used to support conclusions?

-Are there concerns about ethical or regulatory requirements being met?

Reviewer #1: The objectives of the study clearly articulated with a clear testable hypothesis stated. The study design is appropriate to address the stated objectives. The population is clearly described and appropriate for the hypothesis being tested and there is a correct statistical analysis used to support conclusions. the sample size (196) is sufficient to ensure adequate power to address the hypothesis being tested : however, the sample could be even bigger for more powerful conclusions. Unfortunate that the sample was of convenience.

There are correct statistical analysis used to support conclusions.

There are no concerns about ethical or regulatory requirements being met.

Reviewer #2: The objective of the study mentioned in different sections of the manuscript is not the same. In the Background section it is stated that the aim of the study was to correlate ZIKV infection of the placenta and umbilical cord blood with teratogenicity, while in the Title the focus is on the evidence of Zika virus circulation in asymptomatic pregnant women and, finally, in the Methods it is on the prevalence of ZIKV infection in pregnant women. There should be a coherence between the different sections and the manuscript should be written accordingly.

Some aspects of the Methods need to be reviewed:

Sample size

Some points related to the sample size need to be clarified. How the expected frequency of ZIKV infection in the study population was taken into account ? What is meant by “number of diseases that caused microcephaly” and what is meant by “diagnosed cases causing fetal microcephaly” ?

Infant Clinical Assessment

It is not informed which pediatric specialists (paediatricians? neuropediatricians ? other specialties?) performed the physical examination, if the same professionals examined all the children and if there was a standard form or medical record. These information are necessary to evaluate to which extent there was information bias and should be added to the Methods and further explored when discussing the strengths and limitations of the study in the Discussion section. 

Laboratory tests

The laboratory techniques were described in details. However there are some limitations in relation to the use of these techniques to diagnose Zika virus infection which may interfere in the interpretation of the results and therefore need to be added to the discussion. As the study population consisted of pregnant women who underwent vaginal birth or Cesarean section it seems that they were tested just in one point in time. How this fact and also the time window of positivity of these tests may have affected the findings of the study? 

Maternal variables

The pregnant women were recruited when they were in active labor or undergoing elective Cesarean sections and reported to the emergency services of the maternities included in the study. Though the presence of symptoms compatible with Zika virus infection was not listed among the exclusion criteria we assume that they were all asymptomatic pregnant women because it is stated in the title of the manuscript. Were these symptoms investigated by the assistance team who performed the obstetric examination or by the research team ? A history of symptoms compatible with Zika virus infection during pregnancy was investigated? Were they asked about testing for Zika virus at some point during pregnancy?

**Results**

-Does the analysis presented match the analysis plan?

-Are the results clearly and completely presented?

-Are the figures (Tables, Images) of sufficient quality for clarity?

Reviewer #1: The analysis presented match the analysis plan. The results are clearly and completely presented. One inconsistency in the number of pregnant women : 197/196- Check in the rapid test section.You should maybe remind the reader about IgM and IgG for Zika, time of persistence/time of appearence etc ... to understand and interpret better the results. Same p28, you don t give the percentage or results for the head size at birth

Regarding the figures, they are of sufficient quality for clarity. You mention two “table 3” and you have no “table 4”p 20.

Reviewer #2: In relation to the results of the laboratory tests was there an overlapping of positivity between the PCR and IgM rsults? 

It is not clear the rationale for testing the association between RT-qPCR results for placenta samples and evaluation of prenatal care quality. 

Considering the small number of RT-PCR positivity in the umbilical cord and placenta is it worth to test the association with newborn variables ? Which would be the probability of detecting a difference if the difference exists, i.e., which would be the power of the test ?

**Conclusions**

-Are the conclusions supported by the data presented?

-Are the limitations of analysis clearly described?

-Do the authors discuss how these data can be helpful to advance our understanding of the topic under study?

-Is public health relevance addressed?

Reviewer #1: Some of the are conclusions supported by the data presented but not all. Need to be much more elaborated. The limitations of analysis are clearly described maybe the 3 limitations “low complexity”, “diversity of the population” and “ low prevalence of ZIKV” can be elaborated a bit more. The authors discuss partially how these data can be helpful to advance our understanding of the topic under study. For me, the public health relevance can be addressed but need more in depth explanation on how this study can be translated into public health measures and real progress for prenatal care.

Reviewer #2: The finding that ZIKV was circulating in the population of Maranhão in the first semester of 2018, as shown by the PCR results in the umbilical cord blood and placenta samples, is an important issue and should be further explored in the Discussion and Conclusions. Was there also evidence from the notification system that transmission was still happening? Are pregnant women with rash still being notified? If yes, are they being tested? Is there any evidence of circulation of ZIKV from other sources of information? What could be said about the transmission of ZIKV in Maranhao nowadays? 

The discussion of the risks related to teenage pregnancy are out of the scope of the study and do not need to be included. Similarly, the comments made on the level of education of the participants are not linked to the ZIKV transmission. On the other hand, the implications that this finding of a relatively good level of education would have on the inference of the results of the study to pregnant women of the source population, or to the general population, are nor explored.

The comparison with the results of Brasil and colleagues (2016) could be enriched if the different epidemiological scenarios in which these studies were conducted is taken into account. Furthermore, it is worth mentioning if the diagnostic tools to identify adverse outcomes in both studies were comparable. There are other studies in the literature, in addition to the study of Noronha and colleagues (2016) and Shing and colleagues (2016), that could support the discussion of the frequency of adverse outcoms in children born to symptomatic and asymptomatic mothers. 

In the limitations of the study it could be added the small sample size of women with laboratory evidence of ZIKAV infection during pregnancy which limits the power of the study to detect morphological and/or psychomotor changes in the newborns. In this way, could this lack of association be included in the conclusion of the study without other comments? 

In the conclusion the authors say that It is important to include ZIKV testing in prenatal care. However they do not make clear on which finding this conclusion was based and for what reason this testing should be part of the prenatal care.

**Editorial and Data Presentation Modifications?**

Reviewer #1: No editorial suggestion

Need to review the abstract as the conclusion is maybe not enough attractive.

Major modifications are needed in the discussion and conclusion to make this article much more relevant in term of Public Health.

Reviewer #2: The manuscript could focus mainly on the finding of the laboratory evidence of the presence of ZIKV infection in asymptomatic pregnant women in this population and what it adds to the understanding of the circulation of ZIKV in the post-epidemic period.

**Summary and General Comments**

Reviewer #1: This study seems quite innovative with important findings that need to be much more emphasized. More work is needed to link the finding with the practical implementation in prenatal care. What this study will change for pregnant women that might be asymptomatic with ZKV ? What could be the consequences ? What are the risks ? etc .... need more insights.

 No concerns about dual publication, research ethics, or publication ethics.

Reviewer #2: As mentioned in the previous items, the objective of the study should be stated clearly and the main objective should be the axis for the presentation and discussion of the results. The comparisons for which the authors did not present a rationale to support could be excluded and the interpretation of the results for which the sample size was too small should take into account this problem. 

Follow a summary of the previous comments: a) Some points related to the sample size need to be clarified; b) more information is needed in relation to the evaluation of the children and its implications in interpreting the results; c) the limitations of the lab techniques to diagnose ZIKV infection, and their use in just one point in time, may affect the interpretation of the results need to be discussed; d) it is worth informing if a history of symptoms compatible with Zika virus infection during pregnancy was investigated and if these women reported to be tested for ZIKV infection previously; e) the conclusion needs to be rewritten and be based on the main findings.

PLOS authors have the option to publish the peer review history of their article (what does this mean?). If published, this will include your full peer review and any attached files.

Reviewer #1: Yes: Dr sophie ALLAIN IOOS

Reviewer #2: No
---

## [Decision Letter · Decision Letter 1]

23 Mar 2021

Dear Dr Nascimento,

Thank you very much for submitting your manuscript "Evidence of Zika virus circulation in asymptomatic pregnant women in Northeast, Brazil" for consideration at PLOS Neglected Tropical Diseases. As with all papers reviewed by the journal, your manuscript was reviewed by members of the editorial board and by several independent reviewers. The reviewers appreciated the attention to an important topic. Based on the reviews, we are likely to accept this manuscript for publication, providing that you modify the manuscript according to the review recommendations. 

Sincerely,

Abdallah M. Samy, PhD

Deputy Editor

Victor Santos

Deputy Editor

Reviewer's Responses to Questions

**Key Review Criteria Required for Acceptance?**

**Methods**

-Are the objectives of the study clearly articulated with a clear testable hypothesis stated?

-Is the study design appropriate to address the stated objectives?

-Is the population clearly described and appropriate for the hypothesis being tested?

-Is the sample size sufficient to ensure adequate power to address the hypothesis being tested?

-Were correct statistical analysis used to support conclusions?

-Are there concerns about ethical or regulatory requirements being met?

Reviewer #2: The authors say that the study investigated the prevalence of Zika infection in asymptomatic pregnant women in Maranhão state, Northeast Brazil. Though the statement is correct it may give the idea that the study was designed to estimate the prevalence of asymptomatic infection in pregnant women in the Maranhao State. In fact, these results could only be inferred to the two health units sampled in the study (internal validity). The extent to which it could be extrapolated to pregnant women in the Maranhão state depends on the similarity or differences between the sample (women studied), the target population (women attending the two maternities) and the population of pregnant women (external validity) in this state. Therefore it would be better to say “…study investigated the prevalence of Zika infection in asymptomatic pregnant women attending two public maternities in Maranhão state, Northeast Brazil’. 

The information in relation to the sample size calculation is still a bit confusing. As it is written in the text of the manuscript , there is an apparent inconsistency between the objective (This study investigated the prevalence of Zika infection in asymptomatic pregnant women …) and the parameter used to estimate the sample size (… the number of participants considered the prevalence of ZIKV infection that caused microcephaly). If the objective was not related to the pregnancy outcome (microcephaly) why was it considered in the sample size estimate? Please make clear in the text if the sample size was estimated based on the prevalence of ZIKV infection or on the prevalence of ZIKV infection that caused microcephaly. If the latter, please explain the reason. The questioning is not on the sample size that was achieved but on the parameters used to calculate it, and if they are adequate to the objective of the study

**Results**

-Does the analysis presented match the analysis plan?

-Are the results clearly and completely presented?

-Are the figures (Tables, Images) of sufficient quality for clarity?

Reviewer #2: No further comments

**Conclusions**

-Are the conclusions supported by the data presented?

-Are the limitations of analysis clearly described?

-Do the authors discuss how these data can be helpful to advance our understanding of the topic under study?

-Is public health relevance addressed?

Reviewer #2: The authors say in the discussion that it is not clear whether placental alterations are related to 

 fetal adverse or neonatal outcomes or if they are associated with maternal–fetal transmission of ZIKV. However they do not recommend any action (systematic review ? meta-analysis ? any other type of pooled analysis of different studies) to elucidate this point. 

On the other hand they recommend to include ZIKV testing in prenatal care in order to better evaluate the newborns. This recommendation is not backed by the findings of the study. Introducing ZIKV testing to prenatal care is very complex and has several implications in relation to the follow-up of the mothers, in relation to the women’s decision to interrupt (or not) the pregnancy, in relation to the conditions that would be offered to the follow-up of the children, etc). Thus it would be better to withdraw this recommendation or discuss it in depth. 

In relation to the lack of association between ZIKV in umbilical cord blood and morphological and/or psychomotor abnormalities in the newborns, as it may be due to the low power of the study (as mentioned in the discussion), this possible explanation needs also to be clear also in the conclusion not to be misunderstood by the readers.

In the third paragraph of the Discussion the positivity for ZIKV infection of of the study of Brasil et al. (2016) are presented but they are not being used to interpret the results of this paper. In the previous version they were compared to the author’s findings but the reason why they differed was not explored. If a link with the author’s results is not made, it is not clear the reason to present them in the discussion.

**Editorial and Data Presentation Modifications?**

Reviewer #2: (No Response)

**Summary and General Comments**

Reviewer #2: I think the article has improved compared to the first version, but some points can be further worked on according to the recommendations made.

PLOS authors have the option to publish the peer review history of their article (what does this mean?). If published, this will include your full peer review and any attached files.

Reviewer #2: No

Figure Files:

Data Requirements:

Reproducibility:

References

---

## [Editor Report · Decision Letter 2]

27 Apr 2021

Dear Dr Nascimento,

We are pleased to inform you that your manuscript 'Evidence of Zika virus circulation in asymptomatic pregnant women in Northeast, Brazil' has been provisionally accepted for publication in PLOS Neglected Tropical Diseases.

Best regards,

Abdallah M. Samy, PhD

Deputy Editor

Victor S. Santos

Deputy Editor

---

## [Editor Report · Acceptance letter]

18 May 2021

Dear Dr Nascimento,

We are delighted to inform you that your manuscript, "Evidence of Zika virus circulation in asymptomatic pregnant women in Northeast, Brazil," has been formally accepted for publication in PLOS Neglected Tropical Diseases.

Best regards,

Shaden Kamhawi

co-Editor-in-Chief

Paul Brindley

co-Editor-in-Chief
